# An Exploration of Environmentally Sustainable Practices Associated with Alternative Grazing Management System Use for Horses, Ponies, Donkeys and Mules in the UK

**DOI:** 10.3390/ani12020151

**Published:** 2022-01-08

**Authors:** Tamzin Furtado, Mollie King, Elizabeth Perkins, Catherine McGowan, Samantha Chubbock, Emmeline Hannelly, Jan Rogers, Gina Pinchbeck

**Affiliations:** 1Department of Livestock and One Health, Institute for Infection, Veterinary and Ecological Sciences, University of Liverpool, Liverpool CH64 7TE, UK; ginap@liverpool.ac.uk; 2School of Veterinary Science, Institute for Infection, Veterinary and Ecological Sciences, University of Liverpool, Liverpool CH64 7TE, UK; hlmking5@liverpool.ac.uk (M.K.); cmcgowan@liverpool.ac.uk (C.M.); 3Institute of Population Health, University of Liverpool, Liverpool L69 3GL, UK; lizp@liverpool.ac.uk; 4World Horse Welfare, Anne Colvin House, Snetterton, Norwich NR16 2LR, UK; samchubbock@worldhorsewelfare.org; 5The British Horse Society (Welfare Dept) Abbey Park, Stareton, Warwickshire CV8 2XZ, UK; emmeline.hannelly@bhs.org.uk; 6The Horse Trust, Slad Ln, Princes Risborough HP27 0PP, UK; Jan@horsetrust.org.uk

**Keywords:** grazing management, equid, equine, horse, environment, sustainability, pasture management

## Abstract

**Simple Summary:**

Equestrian land could be a potentially important environmental resource, given that pastureland can help to sequester carbon from the atmosphere, prevent soil erosion and provide diverse ecosystems for native plant and wildlife species. However, equestrian land has been overlooked in environmental research and policy. This study reports on the ways which horse, pony, donkey and mule keepers in the UK described environmental practices as part of their equid care. Through an analysis of survey responses (*N* = 758) from equid keepers using alternative grazing systems, we report on three very different management approaches which resulted in keepers aiming to promote healthy pastures and healthy animals. This study provides the basis for future research exploring attitudes to sustainability in equid keepers, as well as evaluating the impact of their efforts.

**Abstract:**

Equestrian grazing management is a poorly researched area, despite potentially significant environmental impacts. This study explored keepers’ use of alternative grazing systems in the care of UK horses, donkeys and mules through an internet survey. The survey was available during the summer of 2020 and comprised closed and open questions, which were analysed with descriptive statistics and iterative thematic analysis, respectively. A total of 758 responses was incorporated into the analysis; the most popular system used were tracks (56.5%), Equicentral (19%), “other” (e.g., non-grass turnout) (12.5%), rewilding (7.5%) and turnout on either moorland (0.7%) or woodland (2.5%). The thematic analysis highlighted that equid keepers across the systems were highly engaged in exploring sustainable practices. Their approaches varied according to each system, yet all aimed to fulfil practices in three major categories, i.e., supporting diverse plant life (usually through restricting equid access to certain areas), supporting wildlife (through the creation of biodiverse environments) and sustainably managing droppings and helminths. Additionally, proponents of the Equicentral systems declared to be aiming to support soil health. These data provide a promising insight into equid keepers’ behaviour and attitudes to sustainability.

## 1. Introduction

The precise amount of land given over to equid use in the United Kingdom (UK) is entirely unknown. However, given that the UK houses at least 847,000 horses and ponies [1,2] and up to 27,500 donkeys [3] and that The British Horse Society (BHS) recommends 1–1.5 acres per animal on permanent grazing [4], it is likely that this is a significant acreage. Therefore, the way in which this land is managed represents a national environmental concern, particularly given the recent environmental strategy by the UK’s Committee for Climate Change, which identifies the embedding of low-carbon farming practices as one of its key recommendations [5]. While land for equestrian-related activity has been largely overlooked in environmental and farming policy and research, the practises used to manage this equestrian land could be impactful in the UK’s move towards overall “greener” practices, both in relation to climate change and conservation of native flora and fauna.

Over 90% of the UK’s horses are turned out in a field at some point each day [6,7]. Traditional horse care involves the use of paddocks, usually with part-time stabling (most usually stabling at night) in order to rest either the horses or the land [6,7]. Horses are most usually grazed on mature grass paddocks and meadow pasture, with grass less than 5 cm in height [6]. Donkeys and mules, often overlooked in equine research, are frequently kept alongside horses on the same pasture, despite differing nutritional and behavioural needs in relation to grazing management [8]. Traditionally, equestrian paddocks are rested seasonally and are often harrowed, rolled and fertilised in this period of rest; however, there has been a lack of research exploring the way in which land used for equids is managed.

The impact of grazing equids on the land depends on the pasture management strategies adopted; high stocking densities, poor weed control and over-grazing can lead to excessive soil damage from treading [9]. This leads to compaction of soil, pugging (wet soil being churned up by heavy footfall, leading to an uneven ground cover) [10] or poaching (where the surface turns into a slurry [11]). Treading can also lead to excessively hard, impervious ground with poor drainage and reduction in plants or bare areas on which only the hardiest plants can grow [11]. This, in combination with the introduction of seeds from supplementary preserved forage, can lead to the overrepresentation of often invasive weed species such as nettles, dock, thistles and buttercups [9]. In these circumstances, soil can become eroded, while nutrient run-off increases [10]; both issues are an environmental concern. One study of cattle, which are of a similar size and weight to horses, found that just one 36 h “treading event” from a herd of cattle during the winter caused subsequent pasture growth to almost halve [12]. Given that horses are often kept on land for an entire season, rather than the 36 h rotations of cattle, as shown in this study, the impact of equids on soil in winter is concerning.

Further, horses are selective grazers, meaning that they choose to eat certain plants and ignore others [13], causing the patchy appearance of many equestrian fields [10,14]. Horses often choose and create “elimination areas” where they defecate and “nutritive areas” where they eat; they prefer not to eat within 1 m of their droppings [15]. As a result, fields which do not have droppings removed can sometimes show extremely uneven growth, with some areas grazed closely and others showing lengthy grasses; taking into account elimination areas and bare soil, horses may eat from less than half the total land available to them [16].

Well-managed equine pastures could provide useful spaces for developing mature pastureland, which could sequester carbon, reduce soil erosion, reduce nutrient run-off and provide rich environments for local insects, bird and mammals [17]. Moderate/light grazing can increase foliage growth overall by stimulating plant production, so optimum amounts of equid grazing can actually contribute to improved forage production [10]. When horses graze lightly on pasture (e.g., at lower stocking densities, or over a larger area), the development of latrine areas is minimised and the risk of parasite infestation may also be lowered [15]. Therefore, exploring the health of equine land and management options could lead to impactful results in relation to improving environmental health and sustainability.

Although this is an under-researched area—there is a dearth of scientific evidence about the relative “health” of equid pastures in the UK—poached areas around gateways, sparse grass cover and weed invasion are thought to be very common [18]. Indeed, only 32.4% of respondents in one study considered the grass cover in their fields “good” [6]. A US study found that horse owners were aware of the options for pasture management, but did not follow recommended stocking densities and management practices [19]; thus far, UK equid-keepers’ awareness of grazing management best practices remains unexplored, but it seems possible that the result might be similar, as reflected in popular opinion [18].

Running alongside these environmental concerns, there are numerous arguments that traditional equid management is no longer suited to equid health. Donkeys, horses, ponies and mules are now most commonly considered “leisure” animals, meaning that they have a very low energy expenditure and are not required to “work” [6]. UK equids are facing an obesity crisis [3,20], partly as a result of grazing inappropriate pastures (for example, diversified dairy pastures), coupled with a lack of exercise [21,22,23]. Anthelmintic resistance as a result of overuse and misuse of anthelmintic products is a serious concern [24,25]. Further, stress and behavioural issues are prevalent [7,26], which has been suggested to be linked with management practices which are incongruent with the needs of today’s leisure horses; for example, a lack of social contact in stable herds [7,27].

Following research into the management of equine obesity, the authors were alerted to creative land management strategies used by UK equid keepers to mitigate the issues outlined above; for example, the use of turnout on low-grass interlinked tracks rather than paddocks [28]. Thus far, these systems have received little attention. Early research into their use has produced mixed results; for example, Hampson et al. (2010) found that pasture layouts, including tracks and spirals, did not yield greater movement in a herd of broodmares, compared with an open paddock [29]; however, pregnant broodmares may not be representative of the movement patterns of non-pregnant mares or geldings, more common in domestic scenarios. Cameron et al. (2021) found that horse keepers favoured track systems for weight loss because of perceived positive welfare for equines with their use [28] and one study on a “dynamic feeding system”, which encouraged movement to access forage, did find effective weight loss as a result [30]. Track systems have also become increasingly popular at equine rescue centres, given their usefulness in providing enriched environments where horses can be managed as a herd, while also managing weight [31].

Given that these systems are becoming increasingly popular in UK equid care, but have not been subject to formal study, a survey was designed to find out more about the non-traditional (hereafter referred to as “alternative”) grazing systems used by keepers of horses, donkeys, ponies and mules (hereafter, equids). This paper reports on an unexpected finding from owner responses to open questions resulting in free-text, that is, that the use of such systems led equid keepers to become invested in supporting the environment, alongside their animal care. Here, we describe the different and sometimes conflicting ways equid owners described aspects of their environmental awareness and activities.

## 2. Materials and Methods

A survey was used to explore alternative grazing system uses by horse, pony, donkey, or mule keepers in the UK. The questionnaire was developed by the research team collaboratively and comprised a mix of categorical, numerical and open-ended questions. The questionnaire aimed to capture the reasons participants chose to use their individual system, their management of the system and the equids within it, the effects on their equids and their perceived benefits of the system. The questionnaire was pilot-tested with five colleagues who used alternative systems; following revisions, it was then uploaded to JISC online survey software (www.jisc.ac.uk, accessed on 11 November 2021). The first page comprised a consent form. The questionnaire is available in the Appendix A. The project was approved by the University of Liverpool’s Veterinary Ethics Committee (number VREC 949).

The survey was released on 10 July 2020 and shared via numerous outlets, including social media (e.g., Facebook pages for proponents of alternative grazing systems) and by the organisations involved in the survey development (e.g., The British Horse Society). Survey responses were closed on 31 August 2020 and the data downloaded for analysis. Two respondents had not completed sufficient information to warrant inclusion and their results were removed, giving a total of 758 participants included in the results.

Descriptive statistics were performed on the quantitative data describing percentage responses with 95% confidence intervals. Chi-squared tests were used to explore whether the relationships between variables were statistically significant. The demographic information is shown in Table 1. The data were analysed using IBM SPSS Statistics 27 (Armonk, NY, USA: IBM Corp).

The open-ended questions were analysed using a three-step inductive thematic analysis [32,33]. This was performed using Microsoft Excel 2019 MSO, (Retrieved from https://office.microsoft.com/excel, 11 November 2021). In this approach, each of the grazing systems were considered individually in the initial stages. Within the data from each system, each open-ended item was initially read through, while notes were made about overriding ideas which were particularly important. Next, each item was considered individually and “codes” were created to reflect the ideas contained within that item. Codes were developed iteratively as they arose in the text and were combined, revised and moved as more data were incorporated. This led to a clearer idea of the important ideas described by the proponents of each system. A count was also taken of the number of occurrences of each theme (for example, the number of participants who mentioned concern over wildlife levels in their fields), to allow a quantitative comparison of themes to be conducted. The results of each system were then compared; for example, the theme “environmental concern” could be compared across systems to determine the different codes and concerns of the users of that system.

The data were then compiled into a report, which was shared with the research group for further questioning and clarification of some areas.

## 3. Results

The emphasis by respondents on bringing in elements of sustainability and environmental concern into their equid keeping was an unexpected result of the survey and is the subject of this paper. A description of the wider results and methods of use of each system is available in Appendix A. Here, we first focus briefly on the participant demographics, before describing the participants’ approaches to environmentally friendly approaches to equid care within each system. The environmental awareness and efforts of the participants were very diverse in terms of their philosophies and practical management and were mainly related to the use of a track system, Equicentral, or rewilding approach to equid care; hence, those are the systems predominantly discussed in this paper. The participants’ responses reflect the unprompted descriptive comments from participants; thus, they cannot be compared statistically. However, their frequency and diversity warrant reporting as a basis for future studies.

### 3.1. Participant Demographics

A total of 758 responses were included in the analysis. The respondents were predominantly in the 55–65 age category and had over 20 years of experience with equids (76.1%; Table 1). Track systems were the most commonly used grazing system (56.5% of responses), followed by Equicentral (19%), “other” (for example, non-grass turnout paddocks or a hybrid of different systems) (12.5%), rewilding (7.5%) and turnout on either moorland areas (0.7%) or areas of woodland (2.5%). Overall, equids were most commonly kept within a herd of 3–5 animals (44.6%), on 2–3 acres of land (26.8%; Table 1).

An in-depth description of how each system worked according to the participants is available (Appendix A); however, we provide a short description of each of the three systems here, for ease of readership (see Table 2).

There was a significant difference in the acreage used across the three systems (χ^2^ *p*-value < 0.0001), with track systems more likely to function on less than 5 acres and rewilding likely to require more than 5 acres (Figure 1). The number of equids kept on the systems was similar across systems (see Figure 2) with 3–5 equids being the most frequent group number; the rewilding system was more likely than the other systems to house groups of more than 5 animals (*p* < 0.01). Combined, these results suggest that track systems operate at the highest stocking densities.

In the three systems, respondents described sometimes using surfaces to cover the ground in some places (for example, road planings, gravel and rubber mats, which sit on top of the mud, hard core, or concrete). All rewilding and Equicentral equids had access to grass, but 8% of track users had entirely removed grass from the track area. On track systems, equids were most likely to receive hay year-round (65% of participants, compared to 51% of Equicentral users and 30% of rewilders), likely due to track systems aiming to provide low-grass environments.

### 3.2. Thematic Analysis of Participants’ Environmental Concerns

The participants described diverse approaches to the sustainable management of equine grazing land, though their approaches usually fell into three thematic categories as follows: supporting an environment for diverse flora; supporting an environment for local wildlife; management of droppings and helminths. The Equicentral system comments yielded an additional theme: supporting soil health. Here, we first describe how and why owners developed their sense of environmental awareness as part of their grazing system use, before describing how users of each system perceived that they were contributing to sustainable equid-keeping practices.

#### 3.2.1. Developing Environmental Awareness

Equid keepers described that they had begun to look for alternative ways of managing their animals, usually as a direct result of health issues with their animals (most commonly laminitis (48.9%; *N* = 372), arthritis (29.6%; *N* = 225) and equine metabolic syndrome (25.9%, *N* = 197). Additionally, some owners described a growing sense of unease with standard practice.

*“Since becoming a horse owner I have been on 3 different livery yards (each for about 5 years). At every one I have watched the grazing visibly deteriorate over that time. Poached in winter, buttercups, docks and much less grass and more weeds, very short stressed grass that wore down teeth and turned to mud with the slightest bit of rain. So I knew there had to be a way of managing that wouldn’t degrade the land, but would still enable a suitable equine environment.”* (Respondent 120, Equicentral user)

Proponents of all systems commonly described aiming to provide a more “healthy” (mentioned 372 times) and “natural” (mentioned 296 times) environment for their animals, compared with traditional paddock/stable set-ups. In each alternative system, respondents aimed to fulfil what they viewed as equids’ ethological needs of “Friends, Forage and Freedom”—they provided environments where their animals could live in a herd of conspecifics (whenever possible), with what the owner felt was appropriate low-energy forage and with freedom of choice and freedom in relation to space. Each system took a different approach to providing those resources.

Users of each type of system frequently described that allowing their horses, ponies, donkeys or mules to live what they viewed as a “natural” lifestyle improved equid agency and wellbeing. This led to changed ownership responsibilities; instead of being a provider of care, the owner was, instead, the provider of an environment in which the animal was perceived as flourishing within an ecosystem. To illustrate this, several owners described their role in relation to being observers of the equid’s interactions with its environment.

*“I used to spend hours doing ‘horse chores’ and my horses were not happy and my land was degrading every year. My land also used to look bad, but now it looks like a proper meadow with lots of wildlife/birdlife, beneficial insects/bees. Keeping my horses in this way has led to me appreciating the environment much more. I now see how horses need to be part of an ecosystem.”* (Respondent 230, Equicentral user)

The change from daily yard work (e.g., mucking out stables) to management of the animal’s immediate environment led some participants to a greater awareness of the local ecosystems and grass “health”. These biodiverse environments were seen as contributing to equid health and interest, as well as being environmentally favourable. Environmental concerns were mentioned by proportionally more respondents from the Equicentral and rewilding systems, compared to track system users (see Table 3).

#### 3.2.2. Track Systems

Track systems were designed to create low-grass environments by keeping animals on a perimeter track. Track users allowed grasses on the central area of the track to grow and these areas experienced a lack of use and low footfall; some owners used this land as additional grazing (mentioned by 20% of track users), or left it to grow as standing or cut hay (mentioned by 6% of track users). Participants referenced this central area as an environmental “haven”, in which diverse species of plant were allowed to flourish for the benefit of the land overall.

*“I don’t cut the grass or make hay. I think that the long grass with seeds left on field is a good way to re nourish field.”* (Respondent 193, track user)

*“The whole site is neutral unimproved species rich grassland; ecological survey shows it holds over 70 species of plants (17 species are grasses).”* (Respondent 384, track user)

One track user had engaged with her local wildlife trust to collaborate in the maintenance of the undisturbed centre portion of her field in summer, which her horses grazed only in winter.

*“The centre of the field is a wildflower meadow of importance locally and so is also managed with this in mind. A seed harvest has been taken from it by the local wildlife trust and on a good year a hay crop.”* (Respondent 704, track user)

Participants discussed a range of approaches to fertilising this central area, ranging from avoiding “chemical” fertilisers altogether (e.g., using harrowed manure), to using low-nitrogen options, or using chemical fertilisers alongside soil analysis. Often, explanations of these choices were related to improving land and plant health without grazing becoming too “rich”.

*“Part of the field was fertisted[sic] with calcified seaweed to improve the root system rather than boost growth.”* (Respondent 318, track user)

*“None of the fields are fertilised or reseeded as we don’t need “rich” grazing, and it’s natural meadow so we don’t want to upset the equilibrium of the land.”* (Respondent 353, track user)

Aside from the central portion of the track, track users were invested in ensuring that their animals had a biodiverse range of plants around the outside of the track, usually for shelter and browsing behaviour.

*“On most of the boundaries we have trees and hedgerows which the horses can utilise for shelter and foraging.”* (Respondent 216, track user)

In some cases, track users had planted a range of horse-friendly hedgerow species around the track (e.g., hawthorn, blackthorn, hazel); others used and maintained pre-existing areas woodland. This was considered to improve the equid’s experience in relation to providing an enriched environment, as well as increasing biodiversity.

##### Supporting an Environment for Local Fauna

As well as supporting diverse plant life, the little-disturbed central portion of the track was described as providing habitats for wildlife and insect life, as part of a developing ecosystem. Nine respondents (2.15% of track users in total; 52% of track users who mentioned environmental concern) described increased wildlife as one of the best things about their track system.

*“The middle is an established area of chalk grassland habitat which supports a lot of wildlife and wildflowers.”* (Respondent 346, track user)

“[the field] *is full of wildlife—pheasant, hare, wild birds etc ... it is a rich habitat.*” (Respondent 193, track user)

Some track users also carried out specific activities designed to encourage or support insect life, a key component in developing dynamic and complex ecosystems.

*“There is an area of cut up logs, now rotting next to where there is a large oak tree in the neighbouring sheep field. These logs have been left to provide scratching posts for legs, and to provide habitat for insects and beasties that like rotting wood.”* (Respondent 101, track user)

*“The track is poo picked daily but the muck is spread around the outer hedgerows or middle of the track as there are lots on dung beetles around.”* (Respondent 687, track user)

##### Management of Droppings and Helminths

Droppings from equids on a track were often confined to a relatively small area and track users emphasised the importance of the daily removal of droppings (mentioned by 11.8%; *N* = 49 track users). This was considered by some to assist in the control of helminths.

*“To help with worm control, I poo pick every day and dispose of it in the allotments. I use worm counting to monitor them but rarely need to use chemical wormers except once a year for tape and encysted worms and bot flies.”* (Respondent 135, track user)

A reduction in the use of anthelmintics was frequently mentioned by track users, perceived to be a result of the frequent removal of droppings. Many users also utilised other animals (19% of track users; *N* = 80) to help manage grazing and particularly with reference to parasite removal through the interruption of species’ life cycles.

*“Sheep are grazed on the system as well as equines. Also guinea fowls are used as they pick up the ticks from the sheep and also will rummage through the horse’s poo and take out anything “nasty” like worm eggs.”* (Respondent 69, track user)

Sheep were the most commonly used co-grazers (mentioned by 64 track users), followed by goats (4 track users), chickens (4 track users), cows (3 track users), alpacas (2 track users) and pigs (1 track user).

##### Environmental Limitations

The main environmental limitation described by users of the track system was the heavy footfall, hence exposed soil, within the tracked area; the majority of users (69.4%; *N* = 286) mentioned the issue of winter mud causing them to remove or alter their tracks in winter. Heavy footfall could also potentially cause further soil degradation.


*“Where the horse constantly walk on the same ground dusty tracks appear and grass doesn’t grow back. Also because they are always grazing short grass the roots are now thin and fragile. I think in a couple of years there will be no grass on the track at all”*


*“Bad for the ground and the sward. Compacts the ground, loafing areas get over trampled. Its far from ideal.”* (Respondent 535, track user)

However, the heavy footfall and “sacrifice area” used for the track could be said to be offset by the fact that the central area is often barely grazed; hence, it was allowed to grow mature and diverse fauna and used as a grazing resource when needed.

#### 3.2.3. Equicentral (Part of Equiculture)

Equicentral users constructed equid health as part of an interdependent balance among soil, plant and animal health. They described their primary concerns as focussing on the soil and land as a means of supporting plant and animal health (8% of Equicentral users; *N* = 11); a concern echoed more frequently and in more depth, than in the other systems:

*“More horse people need to understand how important good land management is when you own horses and land. The land is not just somewhere to turn horses out. Look after the land and it will look after the animals that live on it. Traditional and even ‘track systems’ do not look beyond the horses. And this is short sighted, because that means they are at best just a band aid. A holistic approach is far better, we have to be good land custodians otherwise we will gradually ruin the land we have.”* (Respondent 230, Equicentral user)

*“This [Equicentral system] is about soil health improving horse health. Healthy soil, equals healthy grass. Long roots, long sward, more fibre, fuller horse. Grass recovers quicker to drought and flood…. So grass needs to be nurtured for as close to year round availability and balance available minerals.”* (Respondent 295, Equicentral user)

Equicentral users frequently used language and concepts from the field of regenerative agriculture, for example, talking about using soil and pastures to sequester carbon, limit soil erosion and maintain a balanced ecosystem.

##### Methods for Supporting Soil Health

In order to achieve optimum soil health, Equicentral users employed various strategies. One of the most important aspects was to avoid mud or bare soil, which was viewed as symptomatic of poorly used land and overly heavy equine usage (for example, using the Equicentral principles to avoid mud was mentioned by 12.5%; *N* = 17 respondents). Bare soil was also considered to release carbon and Equicentral users aimed to promote carbon sequestration as part of their principles of field management. Therefore, an ideal Equicentral system had soil which was densely covered with a carpet of mature and biodiverse grasses and plant life.

*“Bare soil = drought = carbon dioxide release. Grass cover = carbon sink. Healthy soils = healthy gut microbiome = healthy manure for soil, so cycle continues.”* (Respondent 188, Equicentral user)

*“I hate mud, ruts and docks. I understand the benefits of a mixed sward of native grass and wildflower species and how important it is to avoid mechanical damage and overgrazing. /Comparing my horse’s field with the eroded soil created by horses on other plots—to the extent the yard manager has asked me several times whether I actually turn my horse out at all!!!”* (Respondent 142, Equicentral user)

As a result, Equicentral users provided equids with free access to a “loafing area”, a surfaced area containing all resources such as shelter, hay, water and enrichment (mentioned by 50.7%; *N* = 69 respondents). Loafing areas were often adapted stable yards, arenas or shelters and their use meant that footfall on fields was limited. Animals could also be kept to loafing areas at certain times (e.g., overnight, or when in particularly wet weather) in order to protect the soil from excessive treading.

*“We use the holding yard as a tool to manage both the health of the horses and the health of the land. Whenever possible we allow access to the grazing. However when the grass gets too short or the land is too wet or dry we reduce the time spent in the field accordingly… When we feel that area needs some recovery time we close the gate to that pasture and open up another. There may be times in very wet or very dry conditions when we have to limit the time spent grazing but this is for the overall health of the pasture. The horses are never stabled but share a shelter within the holding area.”* (Respondent 117, Equicentral user)

As well as fertilising with rotted manure, Equicentral users also described using tactics such as “mulching” (placing old hay or plant waste onto grazing or onto bare soil; mentioned by 4.4%; *N* = 6 users) to improve grazing, by leaving this to break down and form an additional layer of soil and plant life.

*“I also use well rotted compost and hay mulching in the autumn when I lay up my grazing paddocks, and I do the same again in the spring to recover the winter paddock.”* (Respondent 77, Equicentral user)

Between these processes, Equicentral users considered that they were “feeding” their soil, hence contributing to overall plant and equid health.

##### Supporting an Environment for Diverse Flora

Equicentral pastures aimed to encourage diverse native plant life and to avoid improved grasses or monocultured grasses so that equids were grazing on what was considered “healthy” forage. However, users described that it could take several years to develop these mature and biodiverse pastures and, in some cases, described re-seeding their fields.

In order to optimise grass growth, Equicentral users unanimously described splitting their acreage into interlinked paddocks and then rotating access to each area regularly, so that each paddock was grazed very lightly in comparison with traditional turnout methods. For example, animals might move fields every few days or weeks, according to grass health. The heuristic recommended by the Equicentral pioneers is that grass should never be grazed at a length of less than 5 cm. When the fields were resting, Equicentral users (36%; *N* = 46) described field maintenance according to plant health, for example, sometimes harrowing, removing unwanted plants, re-seeding topping, or muck spreading.

As with the track systems, respondents described a range of behaviours in relation to fertiliser use. For example, one described fertilising land as “vital”, while others avoided applying fertilisers altogether.

*“I also use no chemicles [sic] or fertilisers allowing natural growth.”* (Respondent 483, Equicentral user)

One respondent mentioned using low-nitrate fertiliser and another used an equine-specific chemical.

##### Supporting an Environment for Local Fauna

The Equicentral’s focus on mature grassland and care over encouraging plant diversity was considered to directly contribute to creating local ecosystems which encouraged insect and animal life.

*“the fields are a mixture of grass and herbs/wanted weeds to encourage biodiversity, encourage insects, birds and wildlife.”* (Respondent 202, Equicentral user)

*“Bonus is butterfly and moths, increased bird life, increased mouse/vole, so breeding barnowls.”* (Respondent 295, Equicentral user)

*“…. The increase year on year in biodiversity and wildlife using the field. That’s not present on rest of the yard, which is overgrazed barren soil.”* (Respondent 188, Equicentral user)

Equicentral users also described to be co-grazing their fields with sheep (27 mentions), cows (3 mentions), chickens (3 mentions), ducks (1 mention), or goats (3 mentions), in order to better support diversity in their developing ecosystems through a variety of different grazing and behavioural habits and a variety of nutrients replaced in the land via diverse droppings.

*“In the winter we have a flock of sheep grazing in the summer pasture for three weeks. Free manure and hopeful that they eat any larvae left!”* (Respondent 135, Equicentral user)

##### Management of Droppings and Helminths

Because of the lightly grazed paddock rotation, some Equicentral users (11%; *N* = 16) described not to be removing droppings from pasture (or only removing droppings at particular times of year), but, instead, to be allowing droppings to support soil health, often with harrowing. Others (9.5%; *N* = 13) did remove droppings from pasture. Droppings from the yard/loafing area were removed and often composted/rotted (sometimes using the Japanese system “Bokashi” for fermenting organic matter) before being spread back on the field areas as fertiliser.

*“I spend less time than I used to picking up manure (I only pick up from the holding yard, the manure in the paddocks is harrowed in after they are moved to the next paddock).”* (Respondent 230, Equicentral user)

Proponents of this system described to be using faecal egg counts rather than routinely using anthelmintic products; they felt that the use of Equicentral principles had supported the reduction in the parasite burden in their animals.


*“My worm counts have decreased over the years from using this system despite everyone telling me they would increase if I harrowed rather than picked manure in the paddocks.”*


*“Worm counts are non existent as well!”* (Respondent 230, Equicentral user)

##### Environmental Limitations

As with the track system, the Equicentral system relies on sacrificing an area of land (in this case, the surfaced yard or loafing area) in order to preserve the remaining pastureland. However, this area was relatively small compared to the pastureland and could be said to be offset by the care placed on supporting soil health and carbon sequestering in other areas.

#### 3.2.4. Wilding/Rewilding (Including Conservation Grazing Approaches)

In the wilding/rewilding system, the equids were described as part of a complex ecosystem, reverting to their supposed ecological niche. The presence of equids on the land was viewed as a way of enhancing the environment and flora, achieved through light grazing and footfall and sometimes alongside the presence of other grazers.

*“We try NOT TO MANAGE in a traditional sense. There is no stabling, no internal fences. We let them do what horses do and since they came here they have all developed very good condition. This is a longer term project with different grazers to slowly change the land from being overgrazed for years to a natural habitat and have enough land to also allow wildlife to thrive.”* (Respondent 228, rewilding user)

Equid keepers frequently suggested that their animals’ health flourished as a result of grazing on plant life maintained in this way.

*“My grazing, would be described as poor, as it is not uniform lush grass and I let the nettles and thistles grow. I remove poisonous plants but the rest provide a good diet, not too rich in sugar, for ponies, one of whom has Cushings.”* (Respondent 620, rewilding user)

A number of rewilding approaches were identified. While eight participants (13% of rewilders) had what might be considered traditional rewilding or conservation grazing approaches (e.g., animals on a very large acreage with little human intervention), the majority (87%; *N* = 49) incorporated rewilding into more managed or traditional equid care; sometimes for convenience or safety and sometimes due to a small acreage. For example, these equid keepers had areas of the land which were given over to rewilding; equids were given only limited access to such land, in order that they could do their “job” within the ecosystem, but without over-grazing or over-poaching the land.

*“Five rescued Shetlands. as part of a mini conservation project (for my retirement hobby) on hill/moor ground... They are restricted to one area in winter since other parts are too steep and dangerous if it is icy. This is 1100 ‘asl [above sea level] and an exposed windswept Highland moor with heavy snow fall and severe low temperatures. I have grown tree blocks which are now mature enough to have been de-fenced and the ponies graze under the trees where there is also heather cover on the ground. The ponies are a content, gentle team who seem happy in their environment.”* (Respondent 481, rewilding user)

Similarly, the amount of human intervention varied between rewilders; some maintained standard equine care and exercise practices (for example, one mentioned still competing her horse) and many discussed providing supplementary feed (19%; *N* = 11) and shelter (51.6%; *N* = 29) within their rewilding care.

*“some elderley [horses] come in at night they let me know being between 3–37 yrs there body language lets me know. if winter gets so bad and the land is awful like last year i try and bring them in have a barn set up with deep straw.”* (Respondent 726, rewilding user)

##### Supporting an Environment for Diverse Flora

The proponents of full or partial rewilding systems suggested that their land was able to grow diverse plant species, with increased wildlife. Users described that their ponies’ grazing was assisting with the conservation of land, through their specific grazing, movement and droppings patterns.

*“We are conserving plant species that are specific to this area and promoting the health of these plant populations using the ponies as conservation grazers.”* (Respondent 154, rewilding user)

Because of the development of this ecosystem, in which grazing animals effectively fertilised the land through their droppings, none of the rewilding respondents described using fertilisers on their land.

However, with diverse plant life, come plants which are often considered undesirable because they take over, or because they are poisonous to equids. None of the keepers in this project employed a “full” rewilding approach, which would advocate leaving those plants in situ; instead, they were usually removed.

*“The whole area is largely left to its own devices. I pull out any poisonous plants (foxgloves, ragwort, etc) by hand when I see them…. I don’t use weed killer, or any chemicals or sprays…. I don’t cut any of it except during the years when the buttercups are prolific, in which case I might use a long-handled scythe to cut them down before they seed. I don’t use any heavy machinery (not even a quad bike) on any of the land.”* (Respondent 458, rewilding user)

##### Supporting an Environment for Local Wildlife

Proponents of the rewilding system were keen to support local wildlife through the provision of increasingly abundant and diverse habitats and foodstuff, leading to the presence of a variety of fauna. Seven users (12%) mentioned increased wildlife as being one of the best things about the rewilding system.

*“We did the wilding area due to red kites, kestrels, buzzards, foxes and badgers using field and wanted to encourage greater wildlife use without impacting on horses health who needs grass mowed or he will be obese.”* (Respondent 575, rewilding user)

*“I feel I am in a unique position to be able to look after the insects, birds and small mammals who are driven out of the surrounding intensively farmed land.”* (Respondent 620, rewilding user)

Other domestic animals were considered an important part of creating the diversity needed within a wilding area; hence, other animals were brought in or encouraged.

*“We also have 3 pigs and 3 highland cattle to graze different plant species and ‘plough the fields, which has changed the growth of plants in summer—we found much less cow grass growing and much more weeds since starting the system.”* (Respondent 18, rewilding user)

Co-grazers mentioned by the respondents included sheep (mentioned by 10 respondents); goats (mentioned by 3); alpacas (mentioned by 3); pigs (mentioned by 1) and cows (mentioned by 3).

##### Management of Droppings and Helminths

Manure was not removed in rewilding areas, because it was seen as an important and integral part of the ecosystem, in that it could fertilise the surrounding areas.

*“I do not poo pick. The ponies have established poo patches which are managed by the wildlife. The pheasants harrow it. Beetles thrive there.”* (Respondent 620, rewilding user)

However, users did remove manure from areas where animals spent most time (e.g., shelters).

*“I don’t pooh [sic] pick any of it except the shelter area and the beechwood which the ponies use in the summer months to shelter from flies, midges and the hot sun.”* (Respondent 458, rewilding user)

Unlike the other systems, none of the rewilding respondents reported using fertilisers of any sort on their land, preferring to use the manure from their paddocks, or from co-grazed animals, as fertiliser.

In relation to helminths, users described using faecal egg counts rather than using anthelmintic products at regular intervals, with favourable results.

*“I’ve had dung samples taken for a few years... the results have always shown nil necessity to worm them.”* (Respondent 423, rewilding user)

One respondent described using an annual “comprehensive” anthelmintic in Autumn and one used a herbal de-worming product alongside faecal egg counts. Additionally, co-grazing with other species was considered to assist in reducing the helminth burden in pasture.

*“I graze sheep over horse pasture to reduce worm count and just top the long grass.”* (Respondent 557, rewilding user)

## 4. Discussion

In studying the use of alternative grazing systems for equids, this project identified a number of ways in which equid keepers believed that they were supporting biodiversity and promoting healthy, sustainable environments as a part of their equid management.

Across all systems, the practices described by the respondents are generally congruent with agricultural research around improving soil and pasture health. In all three systems, varied species and mature grasses were considered ideal for developing a biodiverse local ecosystem, which would support the retention of carbon in soil, nitrogen fixation and provision of habitats for local fauna [10,36]. The respondents showed an awareness that mature plant cover and light grazing could reduce compaction and soil erosion and understood that “horse-sick” pastures were an environmental concern and a result of pasture misuse or overuse. Proponents of these systems described a preference for organic pastures and, when fertiliser use was described, it was usually with care, for example, alongside soil analysis and, often, of the more environmentally friendly variety (e.g., low-nitrogen; seaweed meal).

The respondents described dropping removal followed by composting and muck-spreading as a sustainable approach to fertilising land and managing manure heaps [37,38]. Removing droppings from pasture is also ideal in terms of disrupting helminth lifecycles, hence reducing the need for the use of anthelmintics [25,39,40], particularly alongside the use of ruminants (e.g., sheep, cows) [25] or other species, which were sometimes employed on all three systems. A previous study found that avoidance of chemicals was a motivating factor in horse owners moving toward faecal egg count use rather than using anthelmintic products at regular intervals [24]; this study supports that finding, given that owners described being similarly motivated in avoiding chemical use for equid and pasture health. Reduced anthelmintic use is also potentially beneficial for invertebrates and particularly dung beetles, which play an important ecological role but can be harmed by the presence of anthelmintics in equid droppings [41,42].

The respondents’ descriptions of environmentally favourable behaviours overlooked some potential benefits of the systems described in this study. For example, across all three systems, the respondents described aiming to reduce or eliminate stabling, which, in turn, reduces the need for commercial bedding products which are required in stables. Given that horses are frequently bedded on imported wood shavings [6], often pine, there is a carbon cost associated with bedding use. Additionally, shavings do not compost as efficiently as other bedding types, such as straw [37].

For each of the three types of grazing system described, there were variations in the precise way in which the land was managed. Variations were related to the type and amount of land being managed, the equids involved and owner-related factors such as ownership of the land, resources and preference. The greatest diversity was shown in the rewilding system. The continuum from giving only limited areas to “wild”, through to set-ups aligned with the more well-known rewilding projects, in which native ponies usually lead a feral lifestyle with minimal human intervention [35,43], encompassed many variations in between. In this project, nearly all rewilders fed supplementary hay and provided shelter and some intervention for their equids. Some researchers argue the case for “rewilding-lite”, adapting rewilding to the practicalities required of farming practices [44]. Similarly, large-scale rewilding projects often allow plants such as ragwort to proliferate due to their role in the ecosystem as a home for specific types of insects [35], despite the fact that they can be poisonous to equids [45]. In this project, the respondents who reported using rewilding practices did report removing these plants, suggesting that equid owners prioritise the health and welfare of their animals if they conflict with the ideals of the rewilding philosophy.

While the equid owners across all three systems described different means of being “environmentally friendly”, the language used by the Equicentral respondents highlighted that they were particularly aware of the link between the soil biome, pasture health and animal health, compared with the users of other systems. This may be a result of the fact that the Equicentral concept has arisen from one specific group, Equiculture, who promotes an education centre, eLearning course and online community which explain the reasons for using the Equicentral system, the practicalities and simplify and translate the environmental science for equid care [46]. Contrastingly, track systems and rewilding are concepts which have become more diluted over time, thus where people can pick and choose resources which might suit them.

However, the track and Equicentral systems relied on some areas being heavily used (the perimeter of fields for tracks; the “loafing area” in an Equicentral system). Some participants were able to surface those areas, describing a range of options including the use of waste products, hardcore (usually comprised of material recycled from the building industry) and various types of “mud mats” (commercial plastic matting designed to sit on top of mud; often made of recycled plastics). However, unsurfaced areas of high footfall (as the perimeter a grass track) could lead to heavily compacted and eroded soil via treading. In this study, track systems operated at the highest stocking densities, higher than the levels recommended by The British Horse Society (1–1.5 acre per horse suggestion for permanent pasture); this could have behavioural impacts, as well as environmental impacts in relation to the compaction and erosion of soil. Further, equids that are shod may cause additional treading and compaction of ground; this study did not collect data on whether animals on these systems were shod or unshod, but it is likely that this could have significant impacts on mud and soil compaction on tracks. Careful management of the central portion of the track may offset some of the environmental limitations of the track perimeter.

This paper focusses on the environmental behaviours of equid owners; however, the equid impact of the use of these systems also warrants attention in further research. As described, many owners used the systems as a way of providing their animals with a “natural” lifestyle in a domestic setting and frequently mentioned their animals’ emotional wellbeing as a result of the provision of “friends, forage and freedom”. Similarly, previous studies have shown track systems to be viewed very positively by horse owners [28]. Nevertheless, issues could occur in some instances; for example, group housing in relatively confined settings (e.g., on a track, or on the “loafing area” or an Equicentral set-up) could potentially lead to resource-guarding issues, particularly if resources are limited, as can be the case with the use of hay feeding stations [47]. Although the owners’ reports of equine behaviour on these systems was generally favourable, ongoing monitoring and observation, as well as further research, would be beneficial.

The study demographics showed a high number of participants who were experienced owners (76.1% had over 20+ years of experience with horses/ponies/donkeys/mules) and predominantly in the 45–64 age bracket (24.3% in 45–54; 26.8% 55–64). This is comparable with other recent, large surveys of horse ownership (for example, a 2020 survey of 5,000 horse owners found an approximately similar age spread and a slightly lower rate of 68% respondents having 20 or more years’ experience [48], but not comparable with regular riders (e.g., the most recent BETA survey, in 2019, found that only 19% respondents were over 45 years (British Equestrian Trade Association (BETA), 2019)). It is unknown whether this demographic spread is indicative of the equid-owning population as a whole. However, it is possible that people who seek alternative grazing management strategies are likely to be older and more experienced, because many respondents commented on dissatisfaction with standard horse care based on their experience over time. Secondly, many respondents commented on the need to own or manage their own land in order to set up an alternative grazing system and land ownership/management may be more common in an older and more experienced cohort. In terms of age and environmental concern, studies report that environmental concern is higher in younger age groups, but environmental behaviours are higher in older generations [49]; therefore, it is possible that the results of this study represent the environmentally directed activities of equid carers who have the capacity to alter their owned or rented grazing land.

This study has some limitations; the survey was intended as an exploratory study about alternative equid management more generally and did not aim to capture information about environmental awareness or environmentally friendly behaviour in equid keepers. This does mean that the data presented here are truly spontaneous; however, the findings cannot be analysed statistically or considered to provide a definitive picture of owner behaviour across the UK. Further research could explore this in more detail, as well as related information such as comparing the number of proportion of alternative grazing system users in the population, compared with standard grazing system use. This has been explored in relation to track systems only, in a study that found that 15.3% of survey respondents used track systems [28]. Further studies could also employ the use of ecological surveys which could assess the actual impact of the system in use. This would also offset a further limitation in that these data are owner-reported and the reported impacts (e.g., increased wildlife and plant biodiversity) may differ from real-life impacts due to human error and confirmation bias from owners who are clearly very passionate about the benefits of their systems.

Another limitation which is relevant for further research is that this survey did not differentiate between donkeys, horses and mules. However, the physiology of these species is such that they have very different environmental and nutritional needs [8,50]. In particular, donkeys have very low nutritional requirements compared to horses; therefore, care would be required to monitor their health and weight when living on Equicentral or rewilding systems where they have access to unlimited forage. Secondly, unlike horses and ponies, donkeys’ coats lack effective waterproofing and winter protection [51,52]; it is essential that they are provided with adequate shelter, whichever system they are kept on. Given that donkeys and mules are sometimes kept alongside horses despite their differing needs [8], the grazing of donkeys and mules on these systems requires additional research.

## 5. Conclusions

This study reveals a positive and promising finding, i.e., equid owners using alternative grazing systems in the UK are engaged in balancing the needs of equids with maintaining a sustainable and ecologically health environment. Given the amount of land managed by equid keepers, encouraging environmentally minded equine care and grazing management is extremely important. Therefore, this study provides the basis for further work on exploring, communicating and encouraging sustainable practice in the equestrian community in the UK and beyond.

## Figures and Tables

**Figure 1 animals-12-00151-f001:**
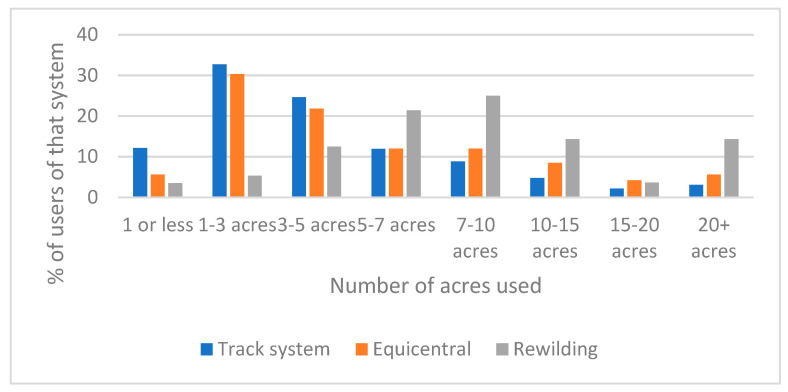
The acreage used by proponents of track, rewilding and Equicentral systems.

**Figure 2 animals-12-00151-f002:**
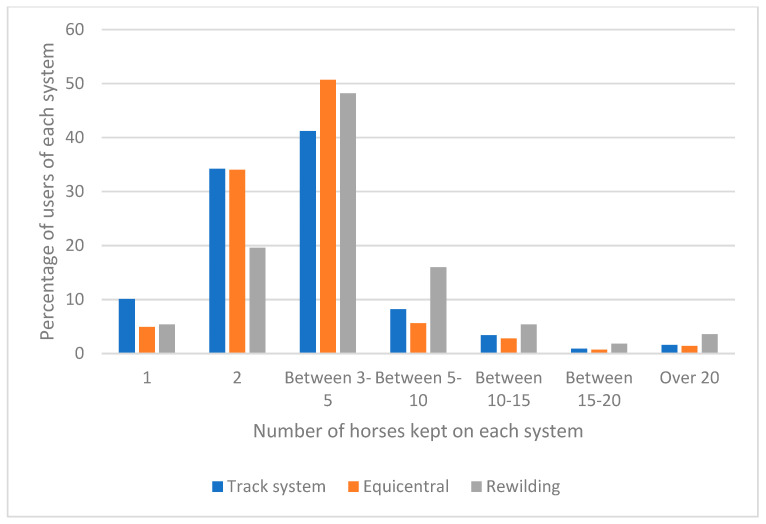
The number of equids kept on track, rewilding and Equicentral systems.

**Table 1 animals-12-00151-t001:** A summary of participant age category, years of involvement with equines and grazing management system use.

Respondent Demographics		Frequency	Percentage (%)	95% Confidence Interval
Age (*N* = 756)	18–25	43	5.7	4, 7.3
	26–34	83	10.9	8.8, 13.3
	35–44	146	19.3	16.6, 22.3
	45–54	184	24.3	21, 27.6
	55–64	203	26.8	23.6, 30.1
	65–74	84	11.1	8.8, 13.5
	75+	13	1.7	0.8, 2.6
Years involved with horses (*N* = 757)	<1	1	0.1	0.0, 0.4
	1–2	3	0.4	0.0, 0.9
	2–5	11	1.5	0.7, 2.4
	5–10	40	5.3	3.7, 7
	10–20	125	16.5	13.9, 19.1
	20+	577	76.1	73, 79.3
Land available for grazing (acres) (*N* = 665)	1 or less	72	9.5	7.4, 11.7
	2–3	203	26.8	23.6, 29.8
	3–5	171	22.6	19.5, 25.6
	5–7	97	12.8	10.4, 15.2
	7–10	83	10.9	8.8, 13.2
	10–15	52	6.9	5.1, 8.6
	15–20	22	2.9	1.7, 4.1
	>20	37	4.9	3.4, 6.5
No of horses/ponies/mules /donkeys kept (*N* = 756)	1	66	0.3	0, 0.7
	2	241	31.8	28.5, 35.1
	3–5	338	44.6	41.2, 48.1
	5–10	64	8.4	6.5, 10.4
	10–15	26	3.4	2.2, 4.7
	15–20	7	0.9	0.3, 1.7
	20+	14	1.8	0.9, 2.9
Type of system (*N* = 754)	Track	428	56.5	53, 60
	Equicentral	144	19	16.4, 21.9
	Rewilding	57	7.5	5.7, 9.5
	Moorland	5	0.7	0.1, 1.3
	Woodland	25	3.30	2.1, 4.6
	Other	95	12.5	10.3, 14.8

**Table 2 animals-12-00151-t002:** A description of each system, showing both the philosophical basis (from the literature) and participants’ descriptions of a typical system.

Type of System	Philosophical Basis	Description
Track system. Seminal text*, Paddock Paradise*, by Jaime Jackson [34]	Horses are evolved to travel long distances each day over varied terrain and graze on low-energy grasses. The track system aims to replicate these factors for domestic horse keeping.	A track is created around the outside of the field and the equids are placed on the track rather than in the central area. Resources (e.g., shelter, water, hay) are then interspersed in different areas of the track to encourage movement. Therefore, for the majority of their time, animals are kept on an area of heavy footfall and low grass; they are most usually fed ad libitum hay whilst on the track. The central area, then, may be cut for hay, strip grazed, or allowed to remain as “standing hay” or “foggage” for winter.
Equicentral (part of Equiculture). Description at www.equiculture.net, accessed on 11 November 2021.	The Equicentral system aims to bring permaculture and sustainable agriculture to horse keeping. Users of the Equicentral system described that their horse care was primarily based around promoting soil health, with the ethos that healthy soil would lead to healthy grasses—hence, healthy horses.	Participants usually described one central area known as the “loafing area”, where equids would find all their resources (shelter, hay, water etc); this area would be large enough for the herd and would be surfaced in order to support year-round use. The equids have access to the fields according to permaculture/mob grazing practices, i.e., the fields are very lightly grazed and never grazed below 5 cm in length. This is purported to encourage the growth of mature-native grasses and to protect the soil, hence providing a host of environmental benefits including the development of ecosystems for native flora and fauna.
Wilding/rewilding/ conservation grazing. Seminal text, *Wilding*, by Isabella Tree [35]	Human management of land disrupts the biodiverse ecosystems of flora and fauna which should be present on land; equids can form an integral part of recreating those diverse ecosystems.	Equids are usually kept on large areas of diverse land (may involve areas of scrub, marsh, woodland and pastureland) and their role is to eat, wander and defecate as a part of the process of recreating diverse ecosystems. In practice, the participants usually described managing some aspects of their care, e.g., feeding, providing shelter and sometimes designating which areas they could use. Rewilding usually allows ecosystems to form naturally, while conservation grazing involves more management or the conservation of certain species.

**Table 3 animals-12-00151-t003:** Number of respondents of each system, who mentioned environmental concerns (for example, soil health, biodiversity, or wildlife) in response to two free-text questions.

	No. Participants Who Mentioned Environmental Concerns in Response to the Question, “What Are the Best Things about the System?”	No. Participants Who Mentioned Environmental Concerns in Response to the Question, “What Were the Reasons for Setting Up the System Initially?”
Track	4% (*N* = 17/417)	2.6% (*N* = 11/419)
Equicentral	22.3% (*N* = 31/139)	23.9% (*N* = 34/142)
Rewilding	27.2% (*N* = 15/55)	37.7% (*N* = 20/53)

## Data Availability

Anonymised data can be shared upon reasonable request.

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
