# Peer review of "An Exploration of Environmentally Sustainable Practices Associated with Alternative Grazing Management System Use for Horses, Ponies, Donkeys and Mules in the UK"

_animals, 2022, doi:10.3390/ani12020151_

Round 1
Reviewer 1 Report
The topic of this article deals with current questions on alternative practices of smallholder farmers for grazing management. Carrying out a large survey on practices is a good approach to collect the main trends in alternative practices. The results present main features of the respondents and the grazing systems used. For the 3 main systems (track, equicentral and wilding), the authors analyse the main environmental concerns, with mentions of some individual answers. Given the size of the sample (over 700 keepers in total), some quantitative results would have been welcome such as the number (or the frequency) of different concerns. The reader has difficulty in relating the qualitative results on concerns presented per grazing system, and the methodology presented in part 2, which proposes to define the main items and their importance in each grazing system, to allow comparison between systems. The discussion focuses first, on the common points observed in horse keepers to promote the environmental approaches then, on specificities of the equicentral system. Thus, as authors propose in conclusion, this article draws up the main trends in alternative practices of the keepers of small of equine herds, and gives arguements to the profession to encourage these practices; but a more in-depth analysis will highlight all the richness of this investigation.
Specific comments
l 45 : “1-1.5 acres per animal, it is likely that this is a significant acreage.”. Could you please clarify what you mean? the value of the recommendation depends seriously on the type of animal (horse, pony or donkey and their physiological needs), the type of pasture, the duration of grazing, etc….
L 78 : “to eat certain plants and ignore others, often causing the patchy appearance of fields” . Yes, but the “patchy appearance” is also related to the grasses height and age.
- Edouard, G. Fleurance, B. Dumont, R. Baumont, P. Duncan, 2009. -Does sward height affect feeding patch choice and voluntary intake in horses? Applied Animal Behaviour Science 119 (2009) – 219-228
L 184 : “Participant demographics” ,and also “structural contexts”, isn’t it?
L 204 : “Combined, these results suggest…”, That would be interesting to have results on the rate of animals per area used in the grazing system.
L 227 :” Equid keepers often described”, could you precise the importance of “often” on the 758 respondents?
L 265 : at this level, a title such as "methods to protect the biodiversity and productivity of the central zone" could have introduced the paragraph.
L 581: “in avoiding chemical use for equid and pasture health.” and specially to protect insects as dung beetle.
L 618 :” Equicentral respondents highlighted that they were particularly aware of the link between the soil biome, pasture health and horse health, compared with the users of other systems.” Yes, perhaps,… but we have no quantification about it, in the part “Results”.
L 631 : however, the part “method” proposes a mode of quantitative analysis of the words, but results of which are difficult to find in part 3
Author Response
The topic of this article deals with current questions on alternative practices of smallholder farmers for grazing management. Carrying out a large survey on practices is a good approach to collect the main trends in alternative practices. The results present main features of the respondents and the grazing systems used. For the 3 main systems (track, equicentral and wilding), the authors analyse the main environmental concerns, with mentions of some individual answers. Given the size of the sample (over 700 keepers in total), some quantitative results would have been welcome such as the number (or the frequency) of different concerns. The reader has difficulty in relating the qualitative results on concerns presented per grazing system, and the methodology presented in part 2, which proposes to define the main items and their importance in each grazing system, to allow comparison between systems. The discussion focuses first, on the common points observed in horse keepers to promote the environmental approaches then, on specificities of the equicentral system. Thus, as authors propose in conclusion, this article draws up the main trends in alternative practices of the keepers of small of equine herds, and gives arguements to the profession to encourage these practices; but a more in-depth analysis will highlight all the richness of this investigation.
Author comment: many thanks for your time and comments; we hope that the changes suggested, which include giving quantified results, will help to facilitate the understanding of the paper.
Specific comments
l 45 : “1-1.5 acres per animal, it is likely that this is a significant acreage.”. Could you please clarify what you mean? the value of the recommendation depends seriously on the type of animal (horse, pony or donkey and their physiological needs), the type of pasture, the duration of grazing, etc….
Author comment: Thank you - you raise a good point here and I agree, but we are only referring to the recommendation made by the organisations. In this instance we do not want to evaluate, critique or discuss that recommendation; the point we make is only that the UK probably uses a lot of land for horses (there is no known figure of exactly how much land is used). While I understand your point, I suggest to leave as is in this instance because we are only making the point that it is likely a lot of land.
L 78 : “to eat certain plants and ignore others, often causing the patchy appearance of fields” . Yes, but the “patchy appearance” is also related to the grasses height and age.
- Edouard, G. Fleurance, B. Dumont, R. Baumont, P. Duncan, 2009. -Does sward height affect feeding patch choice and voluntary intake in horses? Applied Animal Behaviour Science 119 (2009) – 219-228
Author comment: Thank you – reference added
L 184 : “Participant demographics” ,and also “structural contexts”, isn’t it?
Author comment: Demographics is standard use in this instance, and we did not collect structural data (e.g. education level, income etc) – so suggest to leave as is
L 204 : “Combined, these results suggest…”, That would be interesting to have results on the rate of animals per area used in the grazing system.
Author comment: You are quite right; however we collected categorical data rather than count, so unfortunately this cannot be achieved. Thank you for the suggestion though
L 227 :” Equid keepers often described”, could you precise the importance of “often” on the 758 respondents?
Author comment: This sentence has been clarified:
Equid keepers described that they had begun to look for alternative ways of managing their horses as a direct result of health issues with their animals (most commonly laminitis (48.9%; N=372), arthritis (29.6%; N=225) and Equine Metabolic Syndrome (25.9%, N=197), as well as a growing sense of unease with standard practice:
L 265 : at this level, a title such as "methods to protect the biodiversity and productivity of the central zone" could have introduced the paragraph.
Author comment: Suggest to leave as is – we do not talk about methods until later and there are subheadings at that point (this section is just introducing the reasons that owners began to think about the environment, hence subtitle “developing environmental awareness”)
L 581: “in avoiding chemical use for equid and pasture health.” and specially to protect insects as dung beetle.
Author comment: Thank you – added sentence:
Reduced anthelmintic use is potentially beneficial for invertebrates, and particularly dung beetles, which play an important ecological role but can be harmed by the presence of anthelmintics in equid droppings40,41. (Manning 2017, Manning 2018)
L 618 :” Equicentral respondents highlighted that they were particularly aware of the link between the soil biome, pasture health and horse health, compared with the users of other systems.” Yes, perhaps,… but we have no quantification about it, in the part “Results”.
Author comment: Amended in results section, hopefully clearer now!
L 631 : however, the part “method” proposes a mode of quantitative analysis of the words, but results of which are difficult to find in part 3
Author comment: Amended in results section, hopefully clearer now!
Reviewer 2 Report
Dear authors, Many thanks for this piece of work which I found of interest in the climate change age. I share with you most of key concepts you reported. However, I feel that the approach you meant to use to carry out the survey substantially lacks of the nutritional hints. Beyond ecological and behavioural aspects, you deal with horses, ponies, donkeys and mule as a whole in some circumstances, but clearly stated that they have different feeding habits. This should be borne in mind, above all when also nutritional aspects strongly linked with their so different digestive physiology is concerned. Meanwhile, I would also like to drive your attention to the different behaviours in front of weeds (which?) between horses and donkeys, for instance. And the amount of crude fibre and protein they can differently bear (and select upon this reason). Another main difference is the natural plant height and the so called grazing levels based on botanical composition of the patch. On the whole I would better expect some considerations on those aspects, though I am perfectly aware that your questionnaire cannot be as detailed as such to acquire information of basic overall management of grazing. As to parasitic load and hygienic conditions, you may also refer to a paper published in 2020 by Scala et al. on Animals. As to the different hygienic aspect and the risk of other contaminants I would suggest to broaden your discussion by referring to Aboling et al, 2016 on Mycotoxin Research. As to the metabolic push on the feeding behaviour of donkeys and horses in the nature I would suggest to refer to Cappai et al., 2017 on Evology and Evolution and Cappai et al., 2020 Journal of Equine Veterinary Science.
Author Response
Dear authors, Many thanks for this piece of work which I found of interest in the climate change age. I share with you most of key concepts you reported. However, I feel that the approach you meant to use to carry out the survey substantially lacks of the nutritional hints. Beyond ecological and behavioural aspects, you deal with horses, ponies, donkeys and mule as a whole in some circumstances, but clearly stated that they have different feeding habits. This should be borne in mind, above all when also nutritional aspects strongly linked with their so different digestive physiology is concerned. Meanwhile, I would also like to drive your attention to the different behaviours in front of weeds (which?) between horses and donkeys, for instance. And the amount of crude fibre and protein they can differently bear (and select upon this reason). Another main difference is the natural plant height and the so called grazing levels based on botanical composition of the patch. On the whole I would better expect some considerations on those aspects, though I am perfectly aware that your questionnaire cannot be as detailed as such to acquire information of basic overall management of grazing.
Author comment: Many thanks for your kind comments on this paper, and time reviewing it.
As regards your comment about further donkey information – we agree and wish we could have added much more detail about donkeys/mules – future research I think! We have included a large paragraph detailing the issues with horse-designed grazing systems for donkeys and mules (p23-24, line 779-790)
As to parasitic load and hygienic conditions, you may also refer to a paper published in 2020 by Scala et al. on Animals.
Author comment: Thank you – this reference has been added in the sentence:
Removing droppings from pasture is also ideal in terms of disrupting helminth lifecycles and hence reducing the need for the use of anthelmintics25,39,40
As to the different hygienic aspect and the risk of other contaminants I would suggest to broaden your discussion by referring to Aboling et al, 2016 on Mycotoxin Research.
Author comment: Thank you for this suggestion. We agree the field of mycotoxin research is extremely interesting and could be relevant; however, given that there is very little literature in relation to horses and wilding, we feel that any discussion would be speculative. Given that mycotoxins were not mentioned by any respondents, we suggest no change; however, we will absolutely take on your comments in relation to our further research in this area.
But9 As to the metabolic push on the feeding behaviour of donkeys and horses in the nature I would suggest to refer to Cappai et al., 2017 on Evology and Evolution and Cappai et al., 2020 Journal of Equine Veterinary Science.
Author comment: Thank you – reference not added in this instance, as we have contacted leading donkey vet/researchers for advice about donkey metabolism and referenced according to their instructions; also the paper suggested here relates to wild donkeys of a breed unlikely to be relevant to the UK domestic population. However, thank you for the suggestion – we read the paper with care.
Reviewer 3 Report
The manuscript entitled « An exploration of environmentally sustainable practices associated with alternative grazing management system use for horses, ponies, donkeys and mules in the UK” provides an interesting view of land use pertaining to equids in the UK. The authors used a survey approach to investigate how horse keepers in the UK mange the grazing of their equids. It revealed that among the survey respondent the same types of management systems were prevalent, and that the motivations for using these systems were similar.
However, I am missing more objective results out of the survey. I understand that the analysis of open-ended questions is difficult. However, the authors could have offered descriptive results for some of their claims, e.g. when stating “most of the users” for a given system, it would be helpful and more credible citing the exact number of respondents and the percentage that represents. I would remove the statements from individual users and report only percentages on the groups the authors have identified by reading the open-ended questions.
One aspect of land use that is neglected in their study is the status of the hooves of the horses. Horses with shoes will be much more destructive to the soil than horses kept bare-hoof, which was not addressed in this article. Size and mass of the equids will also have an effect on soil compaction. The authors might not have the information on this, but it should at least be in the discussion of the limitations of this work.
Furthermore, I am missing a comparative discussion of the systems, especially between trail and equicentral, and the concentration of resources (food and shelter) in smaller spaces, potentially creating conflicts between animals of a group. This should be added in the discussion.
Overall, I think the subject has many merits, and should be published, but more work is needed on the presentation of these results. Therefore, I recommend to publish, but only after major revisions.
Detailed comments:
Page 2, line 55: the sentence is not exactly scientific, please reword.
Page 2, line 59: I would simplify “Traditionally, equestrian paddocks are rested seasonally, ….”
Page 2, line 69: “with the introduction”
Page 2, line 70: I am not convinced dominance is the right word here. “overrepresentation” might fit better
Page 2, line 70: “such as nettles”
Page 2, line 85: “reducing soil erosion and nutrient run-off, as well as providing …”
Page 2, line 95-97: the sentence is not exactly scientific, please reword.
Page 3, line 103: comma after concerns
Page 3, line 121: no comma after et al.
Page 3, line 133: remove that after “ways”
Page 3, line 134: what do you mean by “support”? For what? From whom?
Page 4, line 137: remove “methodology”. “A survey was used to…”
Page 4, line 150: involved in what?
Page 4, line 155: I would start a new sentence after “confidence intervals”.
Page 4, line 157: “Table” instead of “table”
Page 4, line 172-174: this belongs in the discussion.
Table 1: probably a layout question, but the table is hard to read because the headers are so close together.
Table 1: please maintain a consistent use of comma and space. It is easier to read with a comma and space. If you want to use only a comma, you can, but you need to do it consistently
Table 2: probably a layout problem, but the words are cut off at odd placed due to hyphenations. Recheck that it all makes sense before publication.
Page 6, line 199: you should use the symbol ê•2
Figure 1 and 2: “other” is missing.
Figure 2: there is still “Axis title” in the figure, and “rewilding” is spelled wrong.
Page 7, line 220-221: it is unfair to state that management of droppings of helminths is sustainable. It leads the reader. Remove “sustainable”.
Page 7, line 227-230: you state “often”, how “often”? here a percentage would really improve the credibility of what you are saying
Page 7, line 231-236: I do not think it is scientifically valid to put responses of
Page 8, line 243: same comment with “frequently described”. How frequently?
Page 8, line 257: remove “to” after “also”
Page 8, line 268: same comment as before. How frequently?
Page 8, line 282-286: could you give percentages about these approaches? How many participants used which approach?
Page 9, line 297: how many track users planted trees?
Page 9, line 301: this should appear as a subchapter 3.2.2.1
Page 9, line 318: this should appear as a subchapter 3.2.2.2
Page 9, line 331: this should appear as a subchapter 3.2.2.3
Page 9, line 332-335: could you give percentages about these reasons?
Page 10, line 347-349: do you have statistical evidence for the increased frequency in concerns?
Page 10, line 361: same comment as before. How frequently?
Page 10, line 364: this should appear as a subchapter 3.2.3.1
Page 10, line 365-399: do you have any way to give percentages of numbers of respondents who said what?
Page 10, line 400: this should appear as a subchapter 3.2.3.2
Page 11, line 409: the heuristic what? There is probably a word missing here.
Page 11, line 411: how often?
Page 11, line 414-420: here, percentages would again really help
Page 11, line 421: this should appear as a subchapter 3.2.3.3
Page 11, line 432: how many users?
Page 11, line 436: this should appear as a subchapter 3.2.3.4
Page 11, line 437-452: here, percentages per type of user would again really help
Page 11, line 453: this should appear as a subchapter 3.2.3.5
Page 12, line 459: the numbering in the subchapter is not correct. Should be 3.2.4
Page 12, line 469: how frequently?
Page 12, line 475: you started well with the percentage of respondents, keep it going the whole paragraph
Page 12, line 499: this should appear as a subchapter 3.2.4.1
Page 12, line 501: how many users?
Page 13, line 520: this should appear as a subchapter 3.2.4.2
Page 13, line 536: this should appear as a subchapter 3.2.4.3
Page 13, line 542: how many users?
Page 13, line 547: how many users?
Page 14, line 577: you only wrote about the use of mixed grazing in the chapter about rewilding. Furthermore, you are not stating how often mixed grazing happens depending on the type of system. That would be a relevant result.
Page 14, line 589-594: you have not written about these surfaces in the results, although that would be very interesting to know.
Page 14, line 608: how many in percentages?
Page 15, line 633/line 638: “further research”
Page 15, line 645: “they” instead of “that”
Author Response
The manuscript entitled « An exploration of environmentally sustainable practices associated with alternative grazing management system use for horses, ponies, donkeys and mules in the UK” provides an interesting view of land use pertaining to equids in the UK. The authors used a survey approach to investigate how horse keepers in the UK mange the grazing of their equids. It revealed that among the survey respondent the same types of management systems were prevalent, and that the motivations for using these systems were similar.
Author: Many thanks for your words and approval.
However, I am missing more objective results out of the survey. I understand that the analysis of open-ended questions is difficult. However, the authors could have offered descriptive results for some of their claims, e.g. when stating “most of the users” for a given system, it would be helpful and more credible citing the exact number of respondents and the percentage that represents. I would remove the statements from individual users and report only percentages on the groups the authors have identified by reading the open-ended questions.
Author: Thank you; we have amended the text throughout to include percentages of respondents wherever appropriate. However, we have not removed the quotes, because those are key to reporting thematic analysis and illustrate the respondents’ experiences, which is what will be of interest and use to readers. For further information on qualitative research reporting, please refer to this study design synopsis: https://beva.onlinelibrary.wiley.com/doi/full/10.1111/evj.13436
One aspect of land use that is neglected in their study is the status of the hooves of the horses. Horses with shoes will be much more destructive to the soil than horses kept bare-hoof, which was not addressed in this article. Size and mass of the equids will also have an effect on soil compaction. The authors might not have the information on this, but it should at least be in the discussion of the limitations of this work.
Author: you are absolutely right, and it was an oversight not to collect those data! We have added this to the discussion.
Further, horses who are shod may cause additional treading and compaction of ground; this study did not collect data on whether horses on these systems were shod or unshod, but it is likely that this could have signficiant impacts on mud and soil compaction on tracks.
Furthermore, I am missing a comparative discussion of the systems, especially between trail and equicentral, and the concentration of resources (food and shelter) in smaller spaces, potentially creating conflicts between animals of a group. This should be added in the discussion.
Author: Thank you. It was not our intention to compare the systems or suggest that any is “better” than any other, because that will depend on how they are set up and as we have shown, there is great variation even between the usage of any one type of system. Also, we have provided a comparison in the report of the systems previously published (that focusses much more the nuts and bolts of the system, than the data reported here). However, I have taken your recommendation to add to the discussion about resource guarding and conflict, and we have added the following:
This paper has focussed on the environmental behaviours of equid owners; however, the equid impact of the use of these systems also warrants attention in further research. As described, many owners used the systems as a way of providing their animals with a “natural” lifestyle in a domestic setting, and frequently mentioned their animals’ emotional wellbeing as a result of the provision of “friends, forage and freedom”. Similarly, previous studies have shown track systems to be viewed very poositively by horse owners28. Nevertheless, issues could occur in some instances; for example, group housing in relatively confined settings (e.g. on a track, or on the “loafing area” or an Equicentral set-up) could potentially lead to resource-guarding issues, particularly if resources are limited, as can be the case with the use of hay feeding stations47. Although owner-reports of equine behaviour on these systems was generally favourable, ongoing monitoring and observation, as well as further research, would be beneficial.
Overall, I think the subject has many merits, and should be published, but more work is needed on the presentation of these results. Therefore, I recommend to publish, but only after major revisions.
Author: Thank you; we have made major revisions and hope that those allay your concerns.
Detailed comments:
Page 2, line 55: the sentence is not exactly scientific, please reword.
Author: Amended to:
Traditional horse care involves the use of paddocks, usually with part-time stabling (most usually stabling at night) in order to rest either the horses or land...
Page 2, line 59: I would simplify “Traditionally, equestrian paddocks are rested seasonally, ….”
Author: Amended as suggested
Page 2, line 69: “with the introduction”
Author: Amended as suggested
Page 2, line 70: I am not convinced dominance is the right word here. “overrepresentation” might fit better
Author: Amended as suggested
Page 2, line 70: “such as nettles”
Author: Amended as suggested
Page 2, line 85: “reducing soil erosion and nutrient run-off, as well as providing …”
Author: Amended as suggested
Page 2, line 95-97: the sentence is not exactly scientific, please reword.
Author: Amended to “…..poached areas around gateways, sparse grass cover and weed invasion are very common17”
Page 3, line 103: comma after concerns
Author: Amended as suggested
Page 3, line 121: no comma after et al.
Author: Amended as suggested
Page 3, line 133: remove that after “ways”
Author: Amended as suggested
Page 3, line 134: what do you mean by “support”? For what? From whom?
Author: Amended to “environmental awareness and activities”
Page 4, line 137: remove “methodology”. “A survey was used to…”
Author: Amended as suggested
Page 4, line 150: involved in what?
Author: Amended to “involved in the survey development”
Page 4, line 155: I would start a new sentence after “confidence intervals”.
Author: Amended as suggested
Page 4, line 157: “Table” instead of “table”
Author: Amended as suggested
Page 4, line 172-174: this belongs in the discussion.
Author: I understand, but we need to explain why the paper is framed particularly around this result – we are not describing the overall results here, but are focussing on this unexpected result. I suggest leaving as is so as to frame the results section
Table 1: probably a layout question, but the table is hard to read because the headers are so close together.
Author: Thank you, we will consult the editors about layout options
Table 1: please maintain a consistent use of comma and space. It is easier to read with a comma and space. If you want to use only a comma, you can, but you need to do it consistently
Author: Amended as suggested
Table 2: probably a layout problem, but the words are cut off at odd placed due to hyphenations. Recheck that it all makes sense before publication.
Author: Thank you, will do
Page 6, line 199: you should use the symbol ê•2
Author: Amended as suggested
Figure 1 and 2: “other” is missing.
Author: “other” was not included
Figure 2: there is still “Axis title” in the figure, and “rewilding” is spelled wrong.
Author: Well spotted, thank you! Amended
Page 7, line 220-221: it is unfair to state that management of droppings of helminths is sustainable. It leads the reader. Remove “sustainable”.
Author: Amended as suggested- here and throughout
Page 7, line 227-230: you state “often”, how “often”? here a percentage would really improve the credibility of what you are saying
Author: In relation to this and all further comments on freq/percentages; we have added these wherever appropriate and possible. Please note the aim of a thematic analysis is not to quantify such approaches, so it is not usual to add them into a thematic analysis. However, I appreciate that it can accentuate the results, so have done so as far as possible.
Page 7, line 231-236: I do not think it is scientifically valid to put responses of
Author: As above
Page 8, line 243: same comment with “frequently described”. How frequently?
Author: As above
Page 8, line 257: remove “to” after “also”
Author: This is direct quote from respondent - left
Page 8, line 268: same comment as before. How frequently?
Author: As above
Page 8, line 282-286: could you give percentages about these approaches? How many participants used which approach?
Author: As above
Page 9, line 297: how many track users planted trees?
Author: As above
Page 9, line 301: this should appear as a subchapter 3.2.2.1
Author: Amended as suggested
Page 9, line 318: this should appear as a subchapter 3.2.2.2
Author: Amended as suggested
Page 9, line 331: this should appear as a subchapter 3.2.2.3
Author: Amended as suggested
Page 9, line 332-335: could you give percentages about these reasons?
Author: As above
Page 10, line 347-349: do you have statistical evidence for the increased frequency in concerns?
Author: As above
Page 10, line 361: same comment as before. How frequently?
Author: As above
Page 10, line 364: this should appear as a subchapter 3.2.3.1
Author: Amended as suggested
Page 10, line 365-399: do you have any way to give percentages of numbers of respondents who said what?
Author: As above
Page 10, line 400: this should appear as a subchapter 3.2.3.2
Author: Amended as suggested
Page 11, line 409: the heuristic what? There is probably a word missing here.
Author: The heuristic is the rule of thumb recommended by the authors, so this is fine – altered to “recommended” to increase readability
Page 11, line 411: how often?
Author: As above
Page 11, line 414-420: here, percentages would again really help
Author: As above
Page 11, line 421: this should appear as a subchapter 3.2.3.3
Author: Amended as suggested
Page 11, line 432: how many users?
Author: As above
Page 11, line 436: this should appear as a subchapter 3.2.3.4
Author: Amended as suggested
Page 11, line 437-452: here, percentages per type of user would again really help
Author: As above
Page 11, line 453: this should appear as a subchapter 3.2.3.5
Author: Amended as suggested
Page 12, line 459: the numbering in the subchapter is not correct. Should be 3.2.4
Author: Amended as suggested
Page 12, line 469: how frequently?
Author: As above
Page 12, line 475: you started well with the percentage of respondents, keep it going the whole paragraph
Author: As above
Page 12, line 499: this should appear as a subchapter 3.2.4.1
Author: Amended as suggested
Page 12, line 501: how many users?
Author: As above
Page 13, line 520: this should appear as a subchapter 3.2.4.2
Author: Amended as suggested
Page 13, line 536: this should appear as a subchapter 3.2.4.3
Author: Amended as suggested
Page 13, line 542: how many users?
Author: As above
Page 13, line 547: how many users?
Author: As above
Page 14, line 577: you only wrote about the use of mixed grazing in the chapter about rewilding. Furthermore, you are not stating how often mixed grazing happens depending on the type of system. That would be a relevant result.
Author: We have added figures about the number of co-grazed animals on each system
Page 14, line 589-594: you have not written about these surfaces in the results, although that would be very interesting to know.
Author: Mentioned on P10 237-239
Page 14, line 608: how many in percentages?
Author: As above
Page 15, line 633/line 638: “further research”
Author: Amended
Page 15, line 645: “they” instead of “that”
Author: Thank you, amended